# Cancer Stem Cells in Head and Neck Squamous Cell Carcinoma: Identification, Characterization and Clinical Implications

**DOI:** 10.3390/cancers11050616

**Published:** 2019-05-02

**Authors:** Claudia Peitzsch, Jacqueline Nathansen, Sebastian I. Schniewind, Franziska Schwarz, Anna Dubrovska

**Affiliations:** 1National Center for Tumor Diseases (NCT), Partner Site Dresden, Germany: German Cancer Research Center (DKFZ), 69120 Heidelberg, Germany; Faculty of Medicine and University Hospital Carl Gustav Carus, Technische Universität Dresden, 01307 Dresden, Germany; and Helmholtz Association/Helmholtz-Zentrum Dresden-Rossendorf (HZDR), 01307 Dresden, Germany; Claudia.Peitzsch@uniklinikum-dresden.de; 2OncoRay—National Center for Radiation Research in Oncology, Faculty of Medicine and University Hospital Carl Gustav Carus, Technische Universität Dresden, 01307 Dresden, Germany; jacqueline.nathansen1@tu-dresden.de (J.N.); i.schniewind@gmail.com (S.I.S.); Franziska.Schwarz@uniklinikum-dresden.de (F.S.); 3German Cancer Consortium (DKTK), Partner site Dresden, 01307 Dresden, Germany; 4Helmholtz-Zentrum Dresden-Rossendorf, Institute of Radiooncology—OncoRay, 01307 Dresden, Germany

**Keywords:** head and neck squamous cell carcinoma, HNSCC, cancer stem cells, therapy resistance, biomarkers

## Abstract

Head and neck squamous cell carcinoma (HNSCC) is the sixth most commonly diagnosed cancer worldwide. Despite advances in the treatment management, locally advanced disease has a poor prognosis, with a 5-year survival rate of approximately 50%. The growth of HNSCC is maintained by a population of cancer stem cells (CSCs) which possess unlimited self-renewal potential and induce tumor regrowth if not completely eliminated by therapy. The population of CSCs is not only a promising target for tumor treatment, but also an important biomarker to identify the patients at risk for therapeutic failure and disease progression. This review aims to provide an overview of the recent pre-clinical and clinical studies on the biology and potential therapeutic implications of HNSCC stem cells.

## 1. Introduction

Squamous cell carcinomas make up the majority of head and neck cancers (HNSCC) which have an incidence of around 600,000 new cases per year worldwide, being responsible for about 1–2% of all cancer deaths [1]. The most important etiologic factors are the exogenous noxae tobacco and alcohol [2] together with the infection with high-risk types of human papillomavirus (HPV) [3]. Common manifestation sites include the lip, oral and nasal cavities, pharynx and larynx. The clinical staging is based on the TNM-classification which describes the size of the primary tumor (T), the involvement of regional lymph nodes (N) and the development of distant metastasis (M). TNM system is primarily used to take therapeutic decisions. Locoregional tumors oftentimes can be surgically removed. However, upon diagnosis many tumors are already in a locally advanced stage, which, particularly in combination with an HPV-DNA-negative status [4], leads to a poor prognosis despite multimodal treatment options including surgery, radio- and chemotherapy. More than 50% of patients with locally advanced HNSCC develop metastases or relapse resulting in survival rates of less than a year [5]. Especially in those patients, biomarkers are urgently needed to predict the treatment response, improve treatment selection and provide disease monitoring for early detection of tumor recurrence. Additionally, early diagnosis should be facilitated to improve patient’s outcome [6]. 

Biomarkers are biological indicators of a tumor, which are based on the biological alterations in malignant cells on the different levels including genomic, transcriptomic, proteomic, epigenomic and metabolic processes. While prognostic biomarkers provide information about patient’s outcome and facilitate diagnosis, predictive biomarkers are used to predict the likelihood of patients’ response to a certain therapy. For HNSCC, amplification of epidermal growth factor receptor (EGFR) and the protein level of its ligand, transforming growth factor alpha (TGFα), expression levels of the anti-apoptotic Bcl2, cyclin D1 and cadherin-1 (CDH1) as well as infection with human papilloma virus (HPV) are described as strong prognosticator for patient survival [7]. In 2015, the precision medicine initiative was started by the National Cancer Institute (NCI) to identify genomic drivers in cancer to develop highly sensitive and selective strategies and to guide clinical decision making [8]. For HNSCC, the inactivation of the tumor suppressor p53 and retinoblastoma (pRB) were identified as the leading molecular traits of uncontrolled cell replication. Moreover, mutations in EGFR-MEK, NOTCH, PI3K/AKT/PTEN signaling pathways are frequently observed and induce an aberrant mitogenic signaling. To select an optimal therapy for HNSCC patients from multiple treatment options, predictive biomarkers are applied. The clinical validation of those markers is tested in prospectively designed randomized control trials (RCT). In HNSCC, the expression of Excision Repair 1, Endonuclease Non-Catalytic Subunit (ERCC1) was published to correlate with cisplatin sensitivity [9], β-tubulin III with taxane sensitivity [10], HPV infection with sensitivity to radiochemotherapy (RTOG0129, TROG02.02, RTOG9003, ECOG1308, RTOG1016 trial) [11,12] and EGFR expression with response to anti-EGFR treatment [13].

Biomarker development aims to select biological indicators with a high level of specificity and sensitivity. These biomarkers have to be clinically relevant to detect minimal residual disease (MRD) upon treatment to reduce relapse rates and metastasis. Preclinical and clinical findings indicate that the so-called cancer stem cells (CSCs) are able to survive chemo- and radiotherapy and dynamically adapt to changing environmental conditions, e.g., hypoxia or lack of nutrients. The underlying molecular mechanisms are not fully understood yet, but it was shown that these cells exhibit an enhanced scavenging of reactive oxygen species (ROS), increased DNA repair capacity, elevated cell survival and reduced apoptosis induction [14,15].

The stem cell theory of cancer suggests that malignant cells within a tumor are heterogeneous in their phenotypical and functional properties including differentiation, self-renewal and tumor-initiation capacities. Evidences for tumor cells with stem cell properties were first provided by Stevens, who traced teratoma origin to stem-like cells [16]. In the late 1990s, Dick et al. applied stem cell-specific markers to purify a tumor initiating population in acute myeloid leukemia [17]. One decade later, this population was also proven to be present in some solid tumors. Before this discovery it was assumed that every single cell within a tumor could gain tumorigenic properties through progressive accumulation of advantageous mutations, i.e., the tumors are initiated and develop stochastically. In contrast, the cancer stem cell hypothesis proposes a hierarchical organization of cells within a tumor with a population of self-renewing cells, the so-called cancer stem cells (CSCs), at the apex of the hierarchy. It was shown experimentally that isolated CSCs have a higher potency to initiate xenograft tumors compared to the bulk tumor cell population. In addition, they formed highly heterogeneous tumors representing the histology of the tumor origin [18]. 

Nowadays, it was shown that both models should be combined to describe tumor development due to the fact that CSC themselves undergo clonal evolution during tumor progression upon external stress, e.g., lack of nutrients, hypoxia, therapeutic pressure etc. As a result, different CSC populations may exist in parallel within the same tumor and their competition fuels tumor progression and therapy resistance. In the last years, it was even shown that malignant cells within a tumor (non-CSCs) can gain stem cell properties and transit from non-CSCs into ‘induced’ CSCs [19]. This experimentally evidenced cellular plasticity may be an explanation and the cause of tumor heterogeneity, a major resistance factor impeding therapy response and leading to a poor patient’s outcome. 

In HNSCC, several putative CSC markers, such as CD44 [20], CD133 [21], CD98 [22], CD10 [23], side population (SP) [24], aldehyde dehydrogenase (ALDH) activity [25] and ZsGreen-cODC (C-terminal sequence that directs degradation of ornithine decarboxylase) [26] were described as indicators of tumor cell populations with enhanced tumorigenic potential and reduced sensitivity to chemo- or radiotherapy. For some of these biomarkers, e.g., for CD44 it was even shown that their expression levels in pretreatment biopsies are correlating with therapy response [27,28], and therefore may be applied as prognostic or predictive biomarker to develop more individualized treatment strategies in the future. But on the other hand, almost none of these markers underwent functional testing according to the recommendations by the American Association for Cancer Research (AACR) defined at the Workshop on Cancer Stem Cells in 2006, e.g., by limiting dilution analysis and serial transplantation in vivo upon prospective purification [29]. The concept of CSCs is of high clinical interest due to increasing evidence that all CSCs and CSC-like populations within a tumor have to be eliminated to improve patient’s outcome and increase cancer cure rates.

In this review we summarize and critically discuss recent discoveries within the HNSCC stem cell field in regards to novel isolation techniques, functional analyses and clinical applications.

## 2. Current Methods for HNSCC Stem Cell Enrichment and Analysis

### 2.1. Marker Based CSC Enrichment

In 1994, Lapidot and colleagues suggested the application of stem cell-related markers for the identification of tumor initiating cells in acute myeloid leukemia, thereby opening up a new field of possibilities for CSC research [17]. Since then, a wide range of markers has been identified for CSCs in various solid tumors, including HNSCC [20,21,22,23,24,25,26,30,31,32,33,34,35,36,37,38,39,40,41] (Table 1 and Figure 1). Most of these markers are cell surface proteins, and therefore they can be detected by antibody-based methods like flow cytometry. In HNSCC, the hyaluronic acid receptor CD44 is one of the most common markers for CSC enrichment [20,32,33]. Still, the CSC-related marker panel is growing, yet including the surface proteins CD133 [21,34], CD98 [22], CD166 [35], CD10 [23], CD271 [36], integrins [30] and c-Met [38,39] (Table 1 and Figure 1). 

At least of equal importance is the identification of CSCs through their elevated aldehyde dehydrogenase (ALDH) level. The ALDH proteins comprise a superfamily of cytosolic enzymes involved in various physiological and pathophysiological processes [40]. The activity of the ALDH1 isoenzymes which can be measured with the Aldefluor assay is markedly upregulated in HNSCC CSCs [25,40,41]. Furthermore, the expression of the stem cell transcriptional factors Oct4, Sox2 and Nanog was also correlated with the CSC phenotype in HNSCC [41,42,43,44]. Due to their intra-nuclear localization, these proteins are not well suited for straightforward enrichment approaches and are rather used for the characterization of putative CSC populations by reporter-gene assay [42] or for analysis of CSCs by immunohisto- or immunocytochemistry [37,43,44]. 

Despite comprehensive efforts to find an optimal marker for CSCs in HNSCC, challenges still remain for marker-based CSC enrichment. An ongoing discussion questions the specificity of markers and their restriction to CSCs [45]. For example, a considerable CD44 expression has also been reported in normal epithelial HNSCC tissue and cell lines [32,46]. Furthermore, CSCs are heterogeneous and the absence of one marker does not necessary mean that cells do not possess stem cell properties [47,48,49]. This inconsistency might, at least in part, be attributed to the plasticity of CSCs leading to dynamic changes in marker expression upon treatment [41,50] that will be discussed in this review. Additionally, the existence of several isoforms of a marker protein with different relevance for HNSCC and CSCs can complicate the analysis. For CD44, different variants might play a distinct biological role, but only CD44 variant isoform CD44v3 is highly expressed in CSCs and correlates with HNSCC progression [51]. 

The specificity of marker-based enrichment can be enhanced by using two or multiple markers instead of one. Double-positive cells have been shown to possess higher in vivo tumorigenicity and resistance to treatment than cells positive for only one marker [42,52,53]. To this end, several combinations of CD44 have been suggested, for example with ALDH activity, CD24, CD133 and c-Met [39,42,52,53,54]. However, to further overcome the limitations of CSC markers in HNSCC, the combination with functionality-based enrichment methods appears to be a pragmatic strategy. 

### 2.2. Marker-Independent CSC Enrichment

The properties of CSCs which cause tumor progression and recurrence can also be employed for their selective isolation (Table 2) [20,25,26,42,55,56,57,58]. First, CSCs are capable of growing under anchorage-independent conditions. When cultured in ultra-low attachment plates in serum-free medium, they form non-adherent 3D-structures called tumor spheres [59,60]. Whereas the sphere-forming cells maintain their stem cell properties over several generations, more differentiated cells die of anoikis [61,62]. Additionally, spheres can also be generated in 3D matrices that resemble the original microenvironment and facilitate drug screening [63]. Several reports showed that CSCs can be successfully enriched from primary HNSCC samples or cell lines using the sphere forming assay [42,55,61]. However, efficient sphere generation from primary samples is not always possible [64]. To provide better growth conditions for epithelial CSCs, Kaseb and colleagues suggested a sequential procedure with an additional enrichment step on stromal feeder layers [32]. 

Another characteristic of some CSC populations is their increased chemo-resistance, which can be partially explained by a high expression of ATP-binding cassette (ABC) transporters. These transporters enable the CSCs to efficiently expel chemotherapeutic drugs [65], as well as DNA dyes like Hoechst 33342 [66]. In HNSCC, enrichment by dye efflux capability generates a tumor cell subpopulation, termed “side population”, with higher in vitro clonogenicity, invasiveness and tumorigenicity compared to non-side population cells [24,67,68].

Furthermore, a low proteasome activity has been linked to the CSC phenotype in HNSCC cell lines [26,56]. Cells with low proteasome activity can be identified by the accumulation of the exogenous fusion protein ZsGreen-cODC. This construct consists of the fluorescent protein ZsGreen fused to the C-terminal degron of murine ornithine decarboxylase (cODC), which promotes the proteasomal degradation of the fusion protein in non-CSCs [26,56,69]. However, the time-consuming generation of ZsGreen-cODC cells is a major drawback of this reporter system. 

Comparing different methods for CSC enrichment in HNSCC, Wilson and colleagues obtained distinct CSC subpopulations with very little overlap in their gene expression alterations [70]. Importantly, the subpopulations also showed a different response to the chemotherapeutic drug Paclitaxel. These findings further highlight the necessity to validate the putative CSC populations for their functional properties using the CSC-specific analyses. 

### 2.3. CSC Analysis

The gold standard to evaluate relevant CSC properties like self-renewal, pluripotency and in vivo tumorigenicity is the serial xenotransplantation in immunocompromised mice [29,71]. As each CSC theoretically possesses the ability to repopulate the whole tumor, CSC-enriched subpopulations should be capable of in vivo tumor initiation even from very low cell numbers. To proof self-renewal capacity, putative CSCs need to retain this enhanced tumorigenicity over a series of tumor resection and re-implantation. Furthermore, the xenograft tumors should resemble the histological features of the original tumor, thereby confirming the pluripotent nature of the putative CSCs. In HNSCC cell lines and primary samples, the xenograft assay has been employed numerus times to verify the stem cell phenotype of different CSC subpopulations, for example CD44^+^ [20,32], ALDH^+^ [25,41], CD44^+^ALDH^+^ [42,72] and side population cells [24,68]. In radiation biology, the number of HNSCC clonogenic tumor cells has been suggested to predict xenograft tumor radiosensitivity [73].

Despite being a valuable tool for CSC analysis, the xenotransplantation assay still has some limitations. Due to serial isolation and re-implantation, the CSCs experience frequent changes of their microenvironment, which plays an important role in maintaining stem cell characteristics in HNSCC [74,75]. Thus, it is conceivable that this influences tumorigenicity [29,72]. Furthermore, tumor progression is mostly analysed at heterotopic engraftment sides, whereas an orthotopic model can facilitate the assessment of metastatic potential [76]. For a detailed analysis of unperturbed tumor growth and metastasis, lineage tracing analysis can be employed. To this end, single cells in chemically induced tumors are marked by the expression of a fluorescent protein [77,78,79]. Lineage tracing in HNSCC revealed the presence of a slow proliferating Bmi1+ tumor cell subpopulation that is involved in the formation of metastases in cervical lymph node and showed high therapy resistance [80]. 

However, in vivo techniques require a relatively high amount of time and effort, rendering them unattractive for high-throughput approaches like drug screening. For this reason, in vitro assays have been developed that allow the assessment of some of the key CSC properties. As mentioned above, CSCs from HNSCC tumors or cell lines can be enriched through their ability to form spheres. Moreover, the formation of secondary spheres after dissociation of primary spheres can be used to demonstrate self-renewal capacity [32,81,82,83]. This makes the spherogenic assay an attractive model to analyse the treatment response of tumor cells enriched for CSC populations, for example the surviving fraction after irradiation [32,81]. However, an increased sphere forming capacity does not necessarily mean that these spherogenic cells are highly capable of in vivo tumor initiation [23,84]. These findings emphasize the demand for in vitro assays that faithfully reproduce the physiological microenvironment of CSCs and therefore reflect their functional properties such as self-renewal, pluripotency and tumorigenicity. In this context, the organoid assay recently moved into focus. Organoids are complex 3D-structures derived from pluripotent cells that resemble the original organ structurally and functionally [85]. For HNSCC and other solid tumors, patient-derived organoids have been shown to recreate and maintain the original tumor heterogeneity including a stable population of CSCs [86,87,88]. Moreover, HNSCC organoids can be used to predict in vivo drug response [87], thereby opening great opportunities for the development of fast and reliable screening methods in personalized medicine. 

Based on the enhanced clonogenic capacity of CSCs, the in vitro limiting dilution assay (LDA) allows the determination of their frequency in a tumor sample or cell line [89,90]. When plated as single cells in low numbers, only the cells with a high self-renewal capacity will be able to form colonies within weeks. In HNSCC cell lines, this assay has been successfully applied using very high dilutions [32]. Automatic approaches as developed by Fedr and colleagues provide the opportunity for high-throughput screenings [91]. Furthermore, the limiting dilution analysis has also been performed in vivo to calculate the frequency of tumor initiating cells from HNSCC primary samples and cell lines [56,92]. 

Additionally, CSCs possess the ability to produce both cells retaining the CSC phenotype and more differentiated progenies [29]. To analyse asymmetric division of CSCs obtained from HNSCC tumors, Keysar and colleagues employed the SORE6-system [42]. This reporter system contains a SOX2/Oct4-response element which stimulates the expression of a fluorescent reporter upon binding of the transcription factors, allowing detailed studies of CSC asymmetric division in real time [93].

Each of the above-described assays for CSC analysis has certain limitation, and combinations of different approaches might be useful to gain a deeper understanding of HNSCC CSCs and to develop more efficient anti-cancer therapies. And, not less important, further studies should take in account a high CSC heterogeneity and plasticity.

## 3. HNSCC Hierarchy, Heterogeneity and Plasticity

### 3.1. Clonal Evolution and HNSCC Heterogeneity

The molecular pathogenesis of HNSCC is associated with a number of genetic and epigenetic changes altering gene expression and contributing to the heterogeneity of CSCs. Molecular analysis of patient-derived HNSCC tissues indicates a high intratumoral heterogeneity determined by clonal evolution of the CSC populations. This evolution is influenced by multiple extrinsic factors such as anti-cancer therapy, changes within the tumor microenvironment, immune cell attack, and lack of nutrition as well as by cell-intrinsic features including metabolic reprogramming, genomic instability and epigenetic alterations. One of the first scientific indications describing the phenomena of clonal evolution for HNSCC was a study published by Jin et al. in 2002 [94]. This study examined the karyotype of a panel of naïve HNSCC tumors from different locations and demonstrated that during tumor progression clonal evolution is driven by acquisition of independent mutational events and clonal expansion of CSCs in parallel [94,95]. Another study by Zhang et al. used whole-genome sequencing of several biopsies from primary HPV+ HNSCC tumors and distant metastases to identify intra-tumor heterogeneity and clonal evolution during the metastatic cascade. This study showed that primary tumors are more heterogeneous and metastases evolve from only few clones [96]. A different approach was applied by Mroz et al. who retrospectively analyzed the genomic intratumoral heterogeneity in HNSCC based on whole exome sequencing data from The Cancer Genome Atlas (TCGA). The high heterogeneity was particularly found in p53-mutated tumors which have permanent apoptosis inhibition and aberrant DNA-damage response. Despite the limitations of the analyzed TCGA data, this study demonstrated that tumor heterogeneity is clinically important and predictive of overall survival in HNSCC patients [97]. 

It is hypothesized that local tumors from high risk patients contain already resistant clones that may be selected during anti-cancer therapy leading to reduced therapy response, early relapse and poor patient outcome. To investigate the clonal evolution experimentally on single cell level the cellular barcode technique can be applied in translational models. To this end, a lentiviral barcode library in combination with a fluorescent red-green-blue panel was used to label the tongue squamous cell carcinoma CAL27 cell line with one specific, genetic barcode per cell. The obtained mixed cell population with individually marked cells was grown as xenograft tumors in immunodeficient mice. The diversity of barcodes was assessed in the primary and in local recurrent tumors after surgery to compare clonal complexity. The results revealed a drop of barcode specific signatures from 9% in the primary tumor to 0.2% in the relapsed tumors. Beside the reduction of clonal complexity, the authors also demonstrated a clonal substitution in recurrent HNSCC tumors which have elevated invasiveness and EMT transition. This study demonstrated that recurrent clones are regulated via the hepatocyte growth factor (HGF)-cMet signaling pathway and are characterized by high expression of the putative HNSCC CSC marker CD10 [98]. These results might suggest that only a minor subpopulation within the CSC pool is able to resist treatment and induces recurrent tumor formation. 

The study from Geißler et al. analyzed the phenotypes of the different HNSCC CSC populations in xenograft mice models. The tumors derived from the metastatic HNSCC cell line Detroit-562 were examined for expression of the epithelial cell marker EGFR, the proliferation marker Ki67, the endothelial marker CD31 and the putative CSC marker ALDH1A1 and CD44 by immunofluorescence analysis. This study found that ALDH1A1^high^CD44^low^EGFR^low^ cells were localized in the central tumor area representing a stationary and quiescent CSC population, while a highly migratory and invasive CSC population with an ALDH1A1^neg^CD44^high^EGFR^high^ phenotype was found at the invasive front. It is not known yet if the stationary and migratory CSC populations can be mutually convertible, and what could be the underlying molecular mechanisms [99].

In addition to the functional, phenotypical and genetic characterization of tumor heterogeneity, in silico analysis can be applied to improve our understanding of clonal evolution. Based on ordinary differential equation (ODE) models systems, biologists might gain additional knowledge regarding the step-wise acquisition of mutational patterns leading to the transformation of normal epithelial cells into a malignant state [95,100]. Based on the landmark paper by Hanahan and Weinberg [101,102] six hallmarks are necessary to drive this process including independence on growth signals, insensitivity to the anti-growth cues, evasion of apoptosis, unlimited replication potential, sustained angiogenesis and invasive features. Within the ODE models these characteristics were reduced to four key parameters including angiogenesis, immortality, genetic instability and enhanced replication [95,100]. 

Other mathematical models aim to simulate the effects of a particular mutation in normal tissue stem cells for deregulating stem cell homeostasis, tumor initiation and elevated cell growth. In particular, these models take into account implications of soluble and cellular interactions within the stem cell niche [103]. This analysis enables to distinguish three types of mutations influencing early stage of tumorigenesis including deregulated proliferation, apoptosis evasion and genetic instability. At the same time the acquired oncogenic mutations in tissue stem cells will lead to their transformation and will result in the reduction of normal stem cell pool through replacement by malignant cells [103]. Beside this stochastic process, deterministic or cell-mediated events also influence tumor growth, metastasis and therapy resistance [104]. Computational models simulating tumor progression according to the CSC hypothesis demonstrated that inheritable traits may be altered not only on the genetic level, but also epigenetically conferring growth advantage and increased fitness to a selected CSC population. Moreover, these calculations demonstrated that CSC populations are highly heterogeneous and dynamically regulated during progression from their long-term dormancy state to an aggressive tumor growth [105]. For HNSCC in particular, HPV infection and multimodality therapy has to be taken into account in addition to other microenvironmental factors. Furthermore, in silico models simulated the direct and indirect interactions between HNSCC cells and endothelial cells via the VEGF-Bcl-2-CXCL8 pathway influencing vascular growth and therapy response [106,107,108,109].

In summary, these data demonstrate that clonal evolution and CSC adaptations during tumorigenesis and upon anti-cancer therapy are key determinants of tumor heterogeneity influencing therapy response and patient’s outcome.

### 3.2. Intrinsic and Induced CSC Plasticity

Tumor heterogeneity including the heterogeneity within the CSC population are major determinants for tumor growth, therapy response and metastasis formation. The CSC populations are plastic in nature meaning malignant cells with non-stem cell phenotype are able to gain stem cell features based on the genetic and epigenetic changes [19,110]. Whether these induced CSCs and original CSCs are functionally similar is still a matter of debate. It is also not clear yet if CSCs arise from mutated normal stem cells or from malignant cells with an unstable genome able to reactivate stem cell-specific gene signatures [111,112]. The concept of induced CSCs was first described in breast cancer by the group of Robert A. Weinberg [113,114]. As for today, a number of external factors which drive cancer cell reprogramming have been described such as environmental conditions (e.g., hypoxia, lack of nutrition), growth factors, immune cell attack or anti-cancer therapy. Previous data from our group demonstrated that a population of radioresistant and tumorigenic ALDH^+^ HNSCC CSCs can be induced by X-ray irradiation [41]. Consistent with these results, Vlashi et al. described an irradiation-induced CSC dedifferentiation in HPV-negative but not in HPV-positive HNSCC cell lines [56]. Similar enrichment of CSCs after treatment has been observed after chemotherapy. Cisplatin treatment enriches the population of Oct-4^+^ and Nanog^+^ cells [115] and the fraction of ALDH^+^CD44^+^ CSCs in subcutaneous xenograft models [50]. Another example of the cancer cell reprogramming is an increase of CSC population under hypoxic conditions. The underlying molecular mechanisms are mainly based on the activation of hypoxia inducible factors (HIF) which in turn up-regulate Oct-4 expression [116,117,118].

CSCs are considered to be a candidate cell population for the metastatic tumor spread. This process is regulated by the epithelial-to-mesenchymal transition (EMT) program, which is also important for the maintenance of CSC phenotype and therapy resistance [110,119,120]. 

In addition to the proposed role of EMT for CSC induction and maintenance, recent studies demonstrated an even more important role of EMT for therapy resistance in pancreatic and lung cancer models [110,121,122]. Experimental data for HNSCC suggest a partial overlap of EMT and CSC signatures. For example, Chen and coauthors reported that the TNF receptor associated factor 6 (TRAF6) regulates the CSC and EMT phenotype in HNSCC via TGF-β, CD44, ALDH1, KLF-1 and SOX-2, and elevated expression of TRAF6 clinically correlates with lymphatic spread and poor prognosis [123]. Another study by Chung et al. identified the nerve growth factor receptor CD271 (NGFR; p75NTR) as a marker for HNSCC population with tumor-initiating and metastatic potential regulated via the Snai2/Slug signaling in vitro and in vivo [124]. However, whether EMT is involved in the generation of CSCs in HNSCC is still a matter of debate, and clinical validation of the current findings would strengthen their interpretation. It is assumed that EMT is mainly induced via extrinsic stimuli upon environmental changes. Experimentally this was shown, for example, for the HNSCC cell lines SCC-25 and Detroit 562. When these tumor cells were cultured with epithelial-mesenchymal crosstalk conditioned media from a cancer cells/fibroblast co-culture, the expression of the DNA repair gene excision repair cross complementation group 1 (ERCC-1) was increased. As a result, this epithelial-mesenchymal crosstalk increased radioresistance in these two HNSCC cell lines [125]. On the other hand, cisplatin-resistant HNSCC cells exhibit an enhanced EMT phenotype, which is induced by CD44 via upregulating ZEB1 and suppressing microRNA-200c [126]. Despite EMT is a tightly regulated process, little is known about the dynamics and stability of this induced EMT phenotypes in HNSCC in vitro and in vivo. Moreover, for some other tumor entities, e.g., breast and lung cancers partial EMT with intermediate epithelial/mesenchymal phenotype was described as a stable phenotype associated with elevated invasive properties of tumor cells [110,127,128]. This hybrid EMT phenotype was not described yet for HNSCC.

Morphological and cellular changes associated with EMT include actin stress fiber and vimentin intermediate filament re-arrangements that lead to a front-rear polarity, loss of cell-to-cell interactions and enhanced interaction with extracellular matrix components (ECM) via β1 and β3 integrins resulting in an enhanced invasiveness of tumor cells. The enhanced tumor-initiating ability and drug resistance depends on the EMT extent and is highest at its intermediate level (partial-EMT, semi-epithelial/mesenchymal stage). In HNSCC, EMT is triggered by loss of E-cadherin and upregulation of its counterpart N-cadherin, a process called “cadherin switching” [129,130,131]. Beside N-cadherin, some other intermediate filament proteins and ECM-associated proteins such as alpha-smooth muscle actin (α-SMA), vimentin and laminin 5 are highly expressed in HNSCC compared to normal tissue, and are associated with an invasive phenotype and therapy resistance, whereas their expression levels correlate with poor prognosis [132,133]. A group of transcription factors including Snail, ZEB and TWIST family orchestrate the gene expression changes associated with EMT phenotype and properties, and their increased expression levels correlate with reduced overall survival of HNSCC patients [134,135,136,137,138,139,140,141]. The EMT inducer Twist1 directly regulates the polycomb complex protein BMI-1, which plays an important role in self-renewal and chemotherapy resistance of HNSCC CSCs [80]. Both proteins cooperate to repress E-cadherin and a tumor suppressor protein p16^INK4a^ in HNSCC patients [142]. The already described acquisition of mesenchymal features of carcinoma cells are not permanent and can be converted back to an epithelial state via the opposite mechanism called mesenchymal-to-epithelial (MET) transition. This process has been often found at metastatic tumor sites and might contribute to metastasis formation. Although a growing body of evidence supports the role of EMT and MET mechanism in metastatic disease, careful in vivo studies are still warranted to understand at which extend these processes are involved in tumor cell invasion and metastatic dissemination in HNSCC patients. According to the current view, the contribution of EMT and MET processes to the metastatic spread might depend on the mutational and epigenetic profile of the cell of origin [143,144].

The plastic nature of malignant epithelial cells is related to a widespread reprogramming of gene expression, which is regulated by epigenetic mechanisms including DNA methylation and histone modifications. In particular, active chromatin marks such as histone 3 lysine 4 tri-methylation (H3K4me3) and H3K acetylation are prominent in the epithelial stage while in the mesenchymal stage repressive chromatin marks such as H3K27me3, H3K9me3 and elevated DNA methylation were found [145]. The most common epigenetic modification in HNSCC is DNA hypermethylation in the promoter region of the cyclin-dependent kinase inhibitor 2A (CDKN2A) gene encoding for tumor suppressor p16^INK4a^ that results in enhanced cellular proliferation [146]. Another epigenetic mechanism influencing the CSCs in HNSCC is histone acetylation impacting on chromatin packaging. It was shown that inhibition of histone deacetylase (HDAC) resulted in a decrease in ALDH^+^ CSC population [147,148].

Metastatic spread in HNSCC patients is relatively rare in comparison to other malignancies, but is associated with poor prognosis [149]. Early clinical detection of minimal residual disease (MRD) and early-stage metastasis formation can be monitored by the so-called circulating tumor cells (CTCs) in the blood stream of HNSCC patients. CTC count can be used as prognostic surrogate marker of overall survival and progression-free survival [150,151,152,153]. The predictive value of CTCs for prognosis of disease progression and for therapy selection is not proven yet. Beside the EpCAM-based CTC isolation, additional markers of CTCs are arising e.g., programmed death ligand (PD-L1) for immunotherapy monitoring to distinguish different CTC populations with distinct phenotypic and functional properties [154]. So far, this was not clinically proven yet.

In summary, HNSCC displays a high inter- and intratumoral heterogeneity. The intratumoral heterogeneity depends on the plasticity of CSCs and is associated with increased invasive potential and therapy resistance. A deeper understanding of the underlying molecular mechanisms may have the potential to improve individualized anti-cancer therapy and HNSCC patients’ survival.

## 4. Clinical Implication of HNSCC Stem Cells

### 4.1. The Role of CSCs for Tumor Growth and Metastatic Spread

Given their self-renewal properties, CSCs are thought to play a major role not only in tumor growth and metastasis formation but also in relapse making CSC-related gene and protein expression a promising biomarker candidate and therapeutic target. According to the cancer stem cell model, only complete eradication of the entire CSC population along with non-CSC populations can lead to a definite tumor control and patient cure [155]. However, CSCs seem to have especially effective mechanisms to counteract and evade therapeutic attempts. As already discussed above, cellular plasticity leads to the changes in CSC phenotype depending on the different microenvironmental factors including therapy. A deeper understanding of the mechanisms that contribute to CSC plasticity is needed to effectively combat CSCs. As for today, most of the studies focused on the analysis of CSCs in HNSCC are comprised of in vitro data or immunohistochemical analysis of pre-therapeutic patient tissue specimens and still lack clinical validation in defined prospective studies. Few retrospective studies correlated the CSC marker expression to local response, patient outcome, metastases and relapse rate. As for today, the CSC marker CD44 is of the highest clinical attention. CD44 is a cell-surface glycoprotein and a receptor for hyaluronic acid (HA) involved in cell-cell interaction, adhesion and migration. The interaction of HA and CD44 was found to promote epidermal growth factor receptor (EGFR)-mediated pathways in different tumor entities, in turn leading to tumor cell proliferation and survival via mitogen-activated protein kinase (MAPK) and phosphoinositide 3-kinase (PI3K)/AKT activation [156]. Within the hypoxic tumor areas CD44 gene expression is induced via hypoxia-inducible factor (HIF)-1α [157]. In agreement with these data, increased CD44 levels were associated with tumor hypoxia in HNSCC patients [27]. The study by Wang et al. demonstrated that expression of CD44 v3, v6, and v10 variant isoforms in HNSCC is differently associated with advanced T stage, regional and distant metastasis and radiation failure. These data suggest an involvement of the putative CSC marker in HNSCC tumor cell proliferation and migration [158]. Other studies also support the correlation between an elevated CD44 expression and metastatic disease [76,159]. The exact mechanisms of the involvement of CSCs in the metastatic spread remain obscure [120]. However, the interaction between CD44 and HA seems to play an important role in promoting cytoskeleton rearrangement through RhoA activation and Ca^2+^ mobilization that increases tumor cell migration and invasion [156]. Other CSC populations also might play a role in this process. A recent study by Kim et al. analyzed HNSCC-patient-derived tissue microarrays and found that high expressions of the interleukin-6 receptor (IL-6R) or co-receptor gp130 correlated with low survival. An interleukin IL-6 secreted by endothelial cells induced EMT in HNSCC xenograft tumors and augmented the invasive capacity of ALDH^high^CD44^high^ CSCs. The motility and survival of these CSCs were reduced upon treatment with the humanized anti-IL-6R antibody (Tocilizumab) leading to slower tumor growth with reduced CSC fraction. This study shows a possible mechanism how endothelial cells might contribute to the CSC regulation and metastatic spread [160]. 

### 4.2. Prognostic Role of CSC-Related Biomarkers

Different clinical, pathological and biological parameters have a prognostic value in HNSCC such as tumor stage, tumor volume, invasive fraction, HPV status, hypoxic fraction, tumor-infiltrating lymphocytes as well as CSC marker expression [4,159,161,162,163,164,165,166,167,168]. During the last years an evidence for prognostic and predictive value of CSC-related biomarkers for HNSCC patient stratification is gradually accumulating [27,140,168,169,170,171,172,173,174,175,176,177,178]. The levels of CSC marker expression were correlated with the HPV-DNA status. A smaller proportion of CSCs in HPV-induced tumors suggests a possible explanation for the improved radiosensitivity in HPV-DNA positive head and neck tumors [56,178]. The most accepted and scientifically consolidated HNSCC CSC-related markers are CD44 and SLC3A2/CD98 [28,170,179]. Linge et al. identified high levels of CD44 mRNA, CD44 protein and SCL3A2 mRNA expression as prognosticators for local recurrence in HNSCC after cisplatin-based postoperative radiochemotherapy (PORT-C). Interestingly, this study associated the elevated CSC marker expression with high tumor hypoxia levels [27]. These findings have been validated for an independent patient cohort with locally advanced HNSCC after PORT-C and for a cohort of HNSCC patients receiving primary radiochemotherapy (RCTx) [159,180]. The CD44 expression levels were found to predict 3-year disease-free survival (DFS) and overall survival (OS) in oropharyngeal squamous cell carcinoma (OPSCC) patients undergoing different therapy regimens [181]. The CD44 levels have also been correlated with radiotherapy response and significantly predict local recurrence in patients with early-stage laryngeal cancers [28] and local recurrence and progression free survival in OPSCC [182]. 

The CD98hc (CD98) protein encoded by the SLC3A2 gene and its heterodimerization partner LAT1 constitute a transmembrane amino acid transporter which is shown to be a critical regulator of CSCs in HNSCC [22]. The increased SLC3A2 mRNA expression levels were associated with poor prognosis in HNSCC patients treated with PORT-C or RCTx [27,159]. Both, CD98hc and LAT1 were recently found to identify a poor prognosis subgroup in patients with locally advanced HNSCC treated with RCTx [168]. These data are consistent with another finding showing that a high percentage of CD98-positive cells correlated with worse overall survival and progression-free survival in HPV-positive HNSCC [178]. The CD98 expression levels were also found to correlate with clinical outcome in advanced hypopharyngeal SCC after surgical treatment [183].

All the above mentioned clinical studies are retrospective trials, but prospective, randomized trials would be necessary to demonstrate clinical relevance of CSC-based biomarker analysis for treatment adaptations and individualized cancer therapy. A few studies are initiated already, for example, one study is testing the soluble CD44 marker in saliva for sensitivity, specificity and potential correlation with patient outcome (NCT03148665). The ECOG-E1302 trial is evaluating the frequency of polymorphisms or mutations in the putative CSC marker ATP-binding cassette, sub-family G member 2 (ABCG2) and c-Met oncogene regarding its predictive potential for survival, time to progression, therapeutic response rate and toxicities.

### 4.3. CSC-Targeted Therapies

Potential targeting of CSCs can be realized in different fashions, including targeting of the CSC-related molecules, interfering with the environment promoting CSC functions or inhibiting molecular pathways critical for the CSC maintenance and survival [184]. A general problem with the CSC-targeted therapy is the possible toxicity toward normal cells as the majority of clinically relevant CSC markers are also expressed in normal tissues. At present, little clinical data concerning CSC-targeted therapies in HNSCC is available. The largest body of evidence has been accumulated toward EGFR-targeted therapies, e.g., Cetuximab [185]. However, no emphasis has been put on exploring the clinical effect of these therapeutic agents on CSCs. 

A number of preclinical studies have aimed at the elimination of CSCs [80,186,187,188,189,190,191,192]. Recently Kerk et al. found a higher expression of 5T4, an oncofetal antigen, in HNSCC stem cells. This study analyzed patient tissue arrays and found a correlation between 5T4 levels and lower overall survival. In a preclinical model, 5T4-inhibitor MEDI0641 reduced the CSC fraction and prevented local recurrence [186]. Another study is reporting a CSC-targeting potential for the c-Met inhibitor PF-2341066 and synergistic effects in combination with chemotherapy in HNSCC patient-derived xenograft models (PDX). CSC elimination was achieved by downregulation of the Wnt/β-catenin signaling pathway via disruption of c-Met and frizzled class receptor 8 (FZD8) interaction [187]. Another promising CSC-specific target in HNSCC is the ALDH protein family which is responsible for detoxifying endogenous aldehydes. ALDH proteins are increased upon Cisplatin treatment and radiotherapy and mediate CSC survival in HNSCC. In particular, the ALDH3A1 isoform was reported to be highly expressed by chemoresistant HNSCC CSCs [188], while the expression level of ALDH1A3 protein correlates with tumor radiosensitivity [41]. Molecular targeting with the small molecule inhibitor Aldi-6 reduced ALDH^+^ cells, decreased tumor burden and sensitized HNSCC cells for cisplatin treatment [188]. There is an ongoing debate regarding the clinical applicability of natural compounds with CSC inhibition properties including 6-Gingerol with Wnt/β-catenin targeting potential, β-carotene which inhibits Oct3/4 as well as curcumin, cyclopamine or genistein targeting hedgehog (HH) and Notch signaling. Several studies have investigated their potential in HNSCC in vitro [193,194,195]. Further preclinical characterization of these medicines is necessary before clinical translation (Table 3).

Furthermore, several preclinical studies aiming to validate the efficiency and toxicity of CSC-targeting agents with focus on CD44, cMet and Wnt signaling targeting agents are summarized in Table 4. A phase I single dose escalation study with the anti-CD44 immunoconjugate bivatuzumab mertansine (SB-408075; huC242-DM1) for patients with advanced HNSCC was initiated in 2014, but no results were published so far. Another phase 2 study is testing the efficiency of the selective oral c-Met inhibitor tivantinib (ARQ 197) in combination with cetuximab for recurrent, metastatic HNSCC. First published results evaluating tumor shrinkage, progression-free survival (PFS) and overall survival (OS) demonstrated complete response in 2.5% of patients and partial response in 5% of patients for the combination of cetuximab and tivantinib, while cetuximab alone showed no complete response and a partial response in 7.9% of patients (*p*-value 0.99). The safety and efficacy of the c-Met inhibitor INC280 and cetuximab in comparison to cetuximab or the anti-EGFR antibody panitumumab were also analyzed in HNSCC patients whose disease progressed after cetuximab or panitumumab treatment (NCT02205398). These studies are terminated, but the results are still missing and further clinical validation is needed.

## 5. Conclusions and Future Directions

Despite the specific biomarkers for CSC identification in HNSCC are still under debate, the clinical importance of this cell population has become more evident during the last years. Running clinical trials have correlated the number of CSCs within a tumor biopsy to clinical response and demonstrated the prognostic and predictive potential of CSC-related biomarkers. CSC-targeted therapies are a promising strategy to sensitize resistant tumor cells, to eliminate residual tumor-initiating cells and increase cancer cure rates. But the potential therapeutic efficiency is often impeded by strong side effects due to the targeting common stem cell signaling mechanisms affecting normal tissue stem cells. Moreover, therapy-induced cellular plasticity in HNSCC was so far only shown in preclinical studies and remains to be clinically proven by using e.g., liquid biopsies for therapy monitoring. To fully identify the underlying molecular mechanisms of tumor heterogeneity and cellular plasticity, novel preclinical models including patient-derived xenografts, orthotopic mouse models and immunocompetent genetically-modified mouse models (GEMM) are warranted. In addition, the application of the state-of-the-art technologies, e.g., single cell sequencing to demonstrate cellular heterogeneity, MALDI imaging mass spectrometry (MALDI-IMS) for analysis of histological adaptations or the assay for transposase accessible chromatin with high-throughput sequencing (ATAC-seq) to analyze chromatin compaction will help to develop new therapeutic avenues for individualized therapy and improved patients’ response.

## Figures and Tables

**Figure 1 cancers-11-00616-f001:**
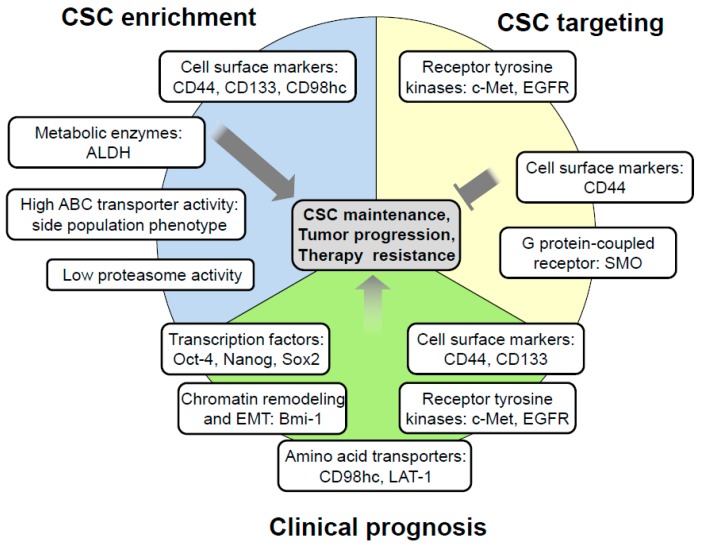
Selected markers for cancer stem cell (CSC) enrichment, targeting and prognostication in head and neck squamous cell carcinoma (HNSCC). Abbreviations: ALDH: Aldehyde dehydrogenases; ABC transporters: ATP-binding cassette transporters; CD98hc: CD98 heavy chain; EGFR: Epidermal growth factor receptor; EMT: Epithelial-mesenchymal transition; LAT1: L-Type Amino acid transporter 1; SMO: Smoothened, Frizzled class receptor.

**Table 1 cancers-11-00616-t001:** Biological role and functional analysis of the HNSCC CSC markers.

Marker	Biological Function	Detection Method	Functional Analysis	Ref.
CD44	Cell surface protein: Receptor for hyaluronic acid, cell-cell and cell-matrix contacts, migration	Flow cytometry, Immunohisto-/Immunocytochemistry	Serial transplantation, xenograft assay	[20]
CD133	Cell surface protein: possible role in membrane organization	Flow cytometry, Immunohisto-/Immnuocytochemistry	Xenograft tumor formation assay	[21]
CD98	Cell surface protein: amino acid transport, integrin signaling	Flow cytometry, Immunohisto-/Immnuocytochemistry	Serial transplanation, xenograft tumor formation assay	[22]
Aldehyde dehydrogenase (ALDH)	Intracellular enzyme oxidizing aldehydes: detoxification, retinoic acid production	Aldefluor assay	Xenotranplantation, Sphere formation assay, (Matrigel/Transwell-invasion assay, Colony formation after irradiation)	[25]
Side population (SP)	ABC transporter-mediated efflux of endogenous and exogenous substances	DNA dye (Hoechst 33342) exclusion assay	Xenograft assay, Serial sphere formation assay, (Matrigel invasion assay, Cell viability after chemotherapeutic drugs)	[24]
ZsGreen-cODC reporter	Low proteasome activity leading to decreased protein degradation	Accumulation of the fluorescent protein ZsGreen fused to the C-terminal degron of murine ornithine decarboxylase (cODC)	Xenograft assay, Sphere formation assay	[26]

**Table 2 cancers-11-00616-t002:** Applications of CSC enrichment methods in HNSCC.

Enrichment Only from Cell Culture	Enrichment from Surgically Excised or Primary Cultured Patient Material	Enrichment from Blood Samples
Low proteasome activity (ZsGreen-cODC reporter) [26,56]	Cell surface proteins (antibody-based methods) [20]	Cell surface proteins (antibody-based methods) [58]
	Aldehyde dehydrogenase (ALDH) activity (Aldefluor assay) [25]	
Stem cell transcription factors (fluorescent reporters) [42]	Sphere formation assay [55]	
	Side population (Hoechst 33342 exclusion assay) [57]	

**Table 3 cancers-11-00616-t003:** CSC-targeting approaches in HNSCC (selected).

Therapeutic Target	Compound	Model System	Results	Reference
5T4 (oncofetal antigen)	MEDI0641	tissue microarray, HNSCC cells (UM-SCC-11B, UM-SCC-22B), patient-derived xenograft (PDX) model	Reduction of CSC fraction, tumor regression	[186]
Bmi1/AP-1	PTC-209	Bmi^CreER^; Rosa^tdTomato^ mice	cisplatin plus PTC-209 potently eradicates Bmi1+ CSCs and inhibits tumor progression	[80]
FGF	BGJ398	HNSCC cell lines	decreased ALDH^high^CD44^high^, sensitization to cisplatin	[189]
Porcupine (PORCN) (Wnt signaling)	LGK974	HNSCC HN30 cell line	High response in HNSCC with Notch loss of function mutation	[190]
ALDH1	Alda-89, Aldi-6	HNSCC cells, xenograft model	Combination with cisplatin enhanced tumor cell death and reduced tumor growth	[188]
CD44	Radionuclide ^186^Re-cmAb (U36)	Patients with Squamous Cell Carcinoma of the Head and Neck	Dose-limiting myelotoxicity, reduction in tumor size	[191]
CD44v6	Anti-CD44v6 antibody BIWA- IRDye800CW and -Indium-111	Fluorescence-guided surgery, locally invasive xenograft model	Detection of tumor regions and invasive zones	[192]
cMET/FZD8	PF-2341066	HNSCC patient-derived xenograft (PDX)	Reduced sphere formation, tumor initiation and metastatic spread	[187]

**Table 4 cancers-11-00616-t004:** CSC-based clinical trials in HNSCC (diagnostics, prognosis and treatment) (selected).

CSC Marker	Pathology/Stage	Treatment	Clinical Endpoint	Sample Size	Trial Identifier	Reference
**1. Diagnostic potential**
Bmi-1	HNSCC, OSCC, LSCC, stage I–IV		DSS, DFS, OS	64–149		[88,140,176]
CD133	HNSCC/LSCC, stage I–IV	OS, DSF	83–98		[174,175]
Oct-4	HNSCC, stage I–V	OS, DSF	60		[173]
Nanog	OSCC	OS	57		[172]
**Sox2**	HNSCC, stage III–IV	Radiation	OS	725		[171]
**2. Prognostic potential**
**CD44**	HNSCC, SCC	RCT	OS, LRC, Distant metastases, Overall survival, Toxicity, Efficiency	29–221	NCT03148665 (OncAlert™), NCT02254018 (Bivatuzumab Mertansine)	[27,169,170]
**cMet**	HNSCC	RCT	LRC, Distant metastases, Overall survival	221		[27]
**CD98**	HNSCC	RCT	LRC, Distant metastases, Overall survival	197-221		[27,168]
**3. Therapeutic potential**
**Hedgehog**	Recurrent HNSCC	IPI-926, Cetuximab	Toxicity, Safety, Efficiency		NCT01255800	
**cMet**		Tivantinib, INC280, Ficlatuzumab	Toxicity, Safety, Efficiency		NCT01619618 (E1302 Trial) NCT01696955 NCT02205398 NCT03422536	
**CD44**		Bivatuzumab Mertansine	Toxicity, Safety, Efficiency		NCT02254018

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
