# Peer review of "Cancer Stem Cells in Head and Neck Squamous Cell Carcinoma: Identification, Characterization and Clinical Implications"

_cancers, 2019, doi:10.3390/cancers11050616_

Round 1

Reviewer 1 Report

This manuscript was well organized and written. It's helpful for people to know and understand the pre-clinical and clinical studies on the biology and potential therapeutic implications of HNSCC stem cells.

Author Response

Reviewer 1

Comments and Suggestions for Authors

This manuscript was well organized and written. It's helpful for people to know and understand the pre-clinical and clinical studies on the biology and potential therapeutic implications of HNSCC stem cells.

Answer: The revised manuscript has incorporated changes based on the reviewer’s suggestions. We are confident that this revision strengthened the manuscript and make it even more relevant for the translational research community.

Reviewer 2 Report

The submitted Review article is a highly significant contribution to the field of Cancer stem cells in head and neck oncology, which provides an overview to the CSC concept and concentrates on clinical and pre-clinical studies on the biological properties of cancer stem cells as well as on their therapeutic relevance. The submitted manuscript covers a intensely needed contribution to this field.

Comments

Please define clearly what is a biomarker in the introduction. Biomarker is supposed to be an indicator of the disease state, as for example the Patient has tumor cells or not, or the therapy is removing Tumor cells or not. Simply said, markers that inform us on the diesase level.  For example prostate specific antigen level in blood is a biomarker  for prostate Cancer or C-reactive protein level in blood is a biomarker for inflammation.  Please summarize literature Information, what biomarkers, if any, are available for head and neck Cancer. For example: human papilloma Virus PCR performed on DNA or circulating tumor cells from blood, and the PCR detection level could be a "biomarker". Please mention such studies in the introduction if you want to write paper on biomarkers. Otherwise you have only predictive markers, if you can predict some Outcome, or prognostic marker, if you Show statistical correlation to prognosis of survival.

Cancer stem cells enrichment methods. Very well written, just one suggestion: please provide a table, which methods could be used only in experimental cell or tissue slice culture conditions, which methods would enrich cancer stem cells from surgically excised tissue and which method has potential for non - invasive cancer stem cells enrichment from blood samples. This table would increase the impact of the manuscript high significantly.

Please provide more clear indication on the relation of EMT with cancer stem cells. Is EMT a process that precedes the formation of cancer stem cells, or the EMT cells and cancer stem cells are independent populations or not? Please take arguments from the available literature on this issue, which is actually a missing information in the fileld.

Author Response

Reviewer 2

Comments and Suggestions for Authors

The submitted Review article is a highly significant contribution to the field of Cancer stem cells in head and neck oncology, which provides an overview to the CSC concept and concentrates on clinical and pre-clinical studies on the biological properties of cancer stem cells as well as on their therapeutic relevance. The submitted manuscript covers a intensely needed contribution to this field.

Comments

Please define clearly what is a biomarker in the introduction. Biomarker is supposed to be an indicator of the disease state, as for example the Patient has tumor cells or not, or the therapy is removing Tumor cells or not. Simply said, markers that inform us on the diesase level.  For example prostate specific antigen level in blood is a biomarker  for prostate Cancer or C-reactive protein level in blood is a biomarker for inflammation.  Please summarize literature Information, what biomarkers, if any, are available for head and neck Cancer. For example: human papilloma Virus PCR performed on DNA or circulating tumor cells from blood, and the PCR detection level could be a "biomarker". Please mention such studies in the introduction if you want to write paper on biomarkers. Otherwise you have only predictive markers, if you can predict some Outcome, or prognostic marker, if you Show statistical correlation to prognosis of survival.

Answer: The definition of biomarker and a description of known biomarkers for HNSCC were added to the introduction of the revised manuscript as follows:

Biomarkers are biological indicators of tumor, which are based on the biological alterations in malignant cells on the different levels including genomic, transcriptomic, proteomic, epigenomic and metabolic processes. While prognostic biomarkers provide information about patient’s outcome and facilitate diagnosis, predictive biomarkers are used to predict the likelihood of patients’ response to a certain therapy. For HNSCC, amplification of epidermal growth factor receptor (EGFR) and the protein level of its ligand, transforming growth factor alpha (TGFα), expression levels of the anti-apoptotic Bcl2, cyclin D1 and cadherin-1 (CDH1) as well as infection with human papilloma virus (HPV) are described as strong prognosticator for patient survival [7]. In 2015, the precision medicine initiative was started by the National Cancer Institute (NCI) to identify genomic drivers in cancer and to develop highly sensitive and selective strategies to guide clinical decision making [8]. For HNSCC, the inactivation of the tumor suppressor p53 and retinoblastoma (pRB) were identified as the leading molecular traits of uncontrolled cell replication. Moreover, mutations in EGFR-MEK, NOTCH, PI3K/AKT/PTEN signaling pathways are frequently observed and induce an aberrant mitogenic signaling. To select an optimal therapy for HNSCC patients from multiple treatment options, predictive biomarkers are applied. The clinical validation of those markers is tested in prospectively designed randomized control trials (RCT). In HNSCC, the expression of Excision Repair 1, Endonuclease Non-Catalytic Subunit (ERCC1) was published to correlate with Cisplatin sensitivity [9], β tubulin III with taxane sensitivity [10], HPV infection with sensitivity to radiochemotherapy (RTOG0129, TROG02.02, RTOG9003, ECOG1308, RTOG1016 trial) [11,12] and EGFR expression with response to anti-EGFR treatment [13].

Cancer stem cells enrichment methods. Very well written, just one suggestion: please provide a table, which methods could be used only in experimental cell or tissue slice culture conditions, which methods would enrich cancer stem cells from surgically excised tissue and which method has potential for non - invasive cancer stem cells enrichment from blood samples. This table would increase the impact of the manuscript high significantly.

Answer: We have added the Table 2 describing the enrichment of the cancer stem cells from cell culture, from surgically excised specimens or primary cultured patient material as well as from blood samples.

Please provide more clear indication on the relation of EMT with cancer stem cells. Is EMT a process that precedes the formation of cancer stem cells, or the EMT cells and cancer stem cells are independent populations or not? Please take arguments from the available literature on this issue, which is actually a missing information in the fileld.

Answer: The description of the relationship between EMT and CSC were added to the section 2.2 “Intrinsic and induced CSC plasticity” of the revised manuscript as follows:

In addition to the proposed role of EMT for CSC induction and maintenance, recent studies indicated even more important role of EMT for therapy resistance in pancreatic and lung cancers [110,121,122]. Experimental data for HNSCC suggest a partial overlap of EMT and CSC signatures. For example, Chen and coauthors reported that the TNF receptor associated factor 6 (TRAF6) regulates the CSC and EMT phenotype in HNSCC via TGF-β, CD44, ALDH1, KLF-1 and SOX-2, and elevated expression of TRAF6 clinically correlates with lymphatic spread and poor prognosis [123]. Another study by Chung et al. identified the nerve growth factor receptor CD271 (NGFR; p75NTR) as a marker for HNSCC population with tumor-initiating and metastatic potential regulated via the Snai2/Slug signaling in vitro and in vivo [124]. However, whether EMT is involved in the generation of CSCs in HNSCC is still a matter of debate, and clinical validation of the current findings would strengthen their interpretation. It is assumed that EMT is mainly induced via extrinsic stimuli upon environmental changes. Experimentally this was shown, for example, for the HNSCC cell lines SCC-25 and Detroit 562. When these tumor cells were cultured with epithelial-mesenchymal crosstalk conditioned media from a cancer cells/fibroblasts co-culture, the expression of the DNA repair gene called excision repair cross complementation group 1 (ERCC-1) was increased. As a results, this epithelial-mesenchymal crosstalk increased radioresistance in these two HNSCC cell lines [125]. On the other hand, cisplatin-resistant HNSCC cells exhibit an enhanced EMT phenotype, which is induced by CD44 via upregulating ZEB1 and suppressing microRNA-200c [126]. Despite EMT is a tightly regulated process, little is known about the dynamics and stability of this induced EMT phenotypes in HNSCC in vitro and in vivo. Moreover, for some other tumor entities, e.g. breast and lung cancers partial EMT with intermediate epithelial/mesenchymal phenotype was described as a stable phenotype associated with elevated invasive properties of tumor cells [110,127,128]. This hybrid EMT phenotype was not described yet for HNSCC.

Reviewer 3 Report

Overall, this paper is well-organized and well-written. It summarizes the results of various original research and makes a valuable contribution to knowledge and understanding of CSCs in HNSCC. Below is the suggestion.

In the third section of “Clinical implication of HNSCC stem cells”, it is recommended to add some results of in vitro and in vivo studies into the subsection of “CSC-targeted therapies” before discussing the clinical trials. Numerous studies have aimed at the elimination of CSCs without harming normal cells. Or it may be beneficial to include a couple of articles regarding the utilization of natural compounds to target CSCs.

Author Response

Reviewer 3

Comments and Suggestions for Authors

Overall, this paper is well-organized and well-written. It summarizes the results of various original research and makes a valuable contribution to knowledge and understanding of CSCs in HNSCC. Below is the suggestion.

In the third section of “Clinical implication of HNSCC stem cells”, it is recommended to add some results of in vitro and in vivo studies into the subsection of “CSC-targeted therapies” before discussing the clinical trials. Numerous studies have aimed at the elimination of CSCs without harming normal cells. Or it may be beneficial to include a couple of articles regarding the utilization of natural compounds to target CSCs.

Answer: We have also added the description preclinical targeting of CSC including treatment with the natural compounds in the section 3.3 of the revised manuscript entitled “CSC-targeted therapies” as follows:

A number of preclinical studies have aimed at the elimination of CSCs [80,186-192]. Recently Kerk et al found a higher expression of 5T4, an oncofetal antigen, in HNSCC CSCs This study analyzed patient tissue arrays and found a correlation between 5T4 levels and lower overall survival. In a preclinical model 5T4-inhibitor MEDI0641 reduced the CSC fraction and prevented local recurrence [186]. Another study is reporting a CSC-targeting potential for the c-Met inhibitor PF-2341066 and synergistic effects in combination with chemotherapy in HNSCC patient-derived xenograft models (PDX). CSC elimination was achieved by downregulation of the Wnt/β-catenin signaling pathway via disruption of c-Met and Frizzled Class Receptor 8 (FZD8) interaction [188]. Another promising CSC-specific target in HNSCC is ALDH protein family which is responsible for detoxifying endogenous aldehydes. ALDH proteins are increased upon Cisplatin treatment and radiotherapy and mediate CSC survival in HNSCC. In particular, the ALDH3A1 isoform was reported to be highly expressed by chemoresistant HNSCC CSCs [187], and the expression levels of ALDH1A3 protein correlates with tumor radiosensitivity [41]. Molecular targeting with the small molecule inhibitor Aldi-6 reduced ALDH+ cells, descreased tumor burden and sensitized HNSCC cells for cisplatin treatment [187]. There is debate regarding the clinical applicability of natural compounds with CSC inhibition properties including 6-Gingerol with Wnt/β-catenin targeting potential, β-carotene which inhibits Oct3/4 as well as curcumin, cyclopamine or genistein targeting hedgehog (HH) and Notch signaling. Several studies have investigated their potential in HNSCC in vitro [193-195]. Further preclinical characterization of these medicines is necessary before clinical translation (Table 3).

We have also added Table 3 providing the examples of the preclinical approaches for CSC targeting.

Round 2

Reviewer 2 Report

Authors have fulfilled the required revision and perfomed a high significant contribution to head and neck oncology.